# Quantum Linear System Algorithm for General Matrices in System Identification

**DOI:** 10.3390/e24070893

**Published:** 2022-06-29

**Authors:** Kai Li, Ming Zhang, Xiaowen Liu, Yong Liu, Hongyi Dai, Yijun Zhang, Chen Dong

**Affiliations:** 1College of Information and Communication, National University of Defense Technology, Xi’an 710006, China; likai17@163.com (K.L.); lxw5054@163.com (X.L.); liuyong09@nudt.edu.cn (Y.L.); gfkd_zyj@163.com (Y.Z.); 2College of Intelligence Science and Technology, National University of Defense Technology, Changsha 410003, China; 3College of Liberal Arts and Sciences, National University of Defense Technology, Changsha 410003, China; hydai@nudt.edu.cn

**Keywords:** system identification, linear systems of equations, quantum algorithm, time complexity

## Abstract

Solving linear systems of equations is one of the most common and basic problems in classical identification systems. Given a coefficient matrix *A* and a vector *b*, the ultimate task is to find the solution *x* such that Ax=b. Based on the technique of the singular value estimation, the paper proposes a modified quantum scheme to obtain the quantum state |x〉 corresponding to the solution of the linear system of equations in $O(\kappa^{2}\sqrt{r}$polylog(mn)/ϵ) time for a general m×n dimensional *A*, which is superior to existing quantum algorithms, where κ is the condition number, *r* is the rank of matrix *A* and ϵ is the precision parameter. Meanwhile, we also design a quantum circuit for the homogeneous linear equations and achieve an exponential improvement. The coefficient matrix *A* in our scheme is a sparsity-independent and non-square matrix, which can be applied in more general situations. Our research provides a universal quantum linear system solver and can enrich the research scope of quantum computation.

## 1. Introduction

System identification [1,2,3] is a common method to determine the mathematical model describing the behavior of classical systems. Thus, the future evolution of the system can be predicted through the identified system model, which is widely applied to common weather forecast, flood forecast, market trend, etc. The traditional system identification method, namely the classical identification method, mainly includes least squares method [4], impulse response method and maximum likelihood method [5,6]. Existing studies [2,3] found that solving linear systems of equations is the basis of system identification problems. In fact, not only system identification problems, the application of linear equations involves various fields of science and engineering, including machine learning [7], partial differential equations [8], classic control system, and so on. Therefore, solving linear systems of equations for general matrices is of great significance.

Due to the importance of linear systems of equations in various fields, the solution of linear equations has become an enduring issue, and many algorithms derived therefrom. The classical solvers mainly include: matrix elimination method [9] and Kaczmarz method [10]. The most famous one of the former is the Gaussian elimination method, which is often used to solve small linear systems of equations and is suitable for a general coefficient matrix. The Kaczmarz method is generally more practical in the field of large-scale linear equations. The running time for these classical solvers scales as $O(n^{3})$, where *n* is the size of the matrix, which will cost a lot of computing resources in solving large-scale linear systems. However, quantum computation [11,12,13] is capable of greatly reducing the time complexity for matrix operation and numerical calculation, which can be regarded as a promising attempt as a computing tool to improve the identification efficiency.

Quantum computation is an emerging computing technology that regulates quantum information units to perform high-efficiency calculations based on the laws of quantum mechanics, including coherent superposition and entanglement [14]. In 1994, Shor proposed the algorithm for prime factorization [15] with exponential acceleration over classical algorithms, which shows the potential of quantum computation for the first time. Since then, quantum computation has reached an era of rapid development. In recent years, scholars have also made significant progress in quantum algorithm research, including Grover algorithm [16], quantum simulation [17,18,19], duality algorithm [20,21,22], linear systems of equations solver [23,24,25], matrix multiplication algorithm [26,27], and so on. For the high-dimensional linear systems of equations, there have been breakthroughs in the field of quantum computation. In 2009, Harrow, Hassidim and Lloyd [23] proposed the quantum linear system algorithm (HHL) to obtain the quantum state $\left|x\rangle =\right|A^{-1}b\rangle$ corresponding to the solution of Ax=b in time O(polylog(n)), where the sparse matrix $A\in R^{n\times n}$ and $x,b\in R^{n}$, which can improve the computational efficiency with an exponential speed-up over classical algorithms. The HHL algorithm is of great significance in the field of quantum information processing and has a wide range of applications in big data, machine learning, numerical computing and other scenarios. In 2018, Wossnig et al. [28] proposed a sparsity-independent quantum linear system algorithm (QLSA) based on a quantum singular value estimation algorithm (QSVE). After that, Shao and Xiang [29] modified the QSVE algorithm to adapt to the non-Hermitian case. Current algorithms for linear systems have been widely applied in the emerging research area of quantum information processing. However, existing quantum algorithms have different restrictions on matrix *A*, such as the most typical one of HHL algorithm, which requires *A* to be a sparse Hermitian matrix so that the unitary transformation $e^{iAt}$ [30,31] can be realized in a constant time. At present, the quantum algorithm suitable for arbitrary linear system of equations has not been fully studied.

Without loss of generality, existing quantum algorithms assumed that the coefficient matrix *A* is Hermitian as it is well known that the general case can be reduced to the Hermitian case by embedding a general rectangular matrix *M* into a block antidiagonal Hermitian matrix with the elements of $M^{†}$ and *M* in the lower and upper half, respectively [28]. Different from previous algorithms, we proposed a modified quantum scheme to solve the cases of general matrices directly, which can reduce the time complexity of solving the linear system of equations. Moreover, it may not be easy to expand *A* into a Hermitian matrix when *A* is given as quantum information. However, our scheme does not need such expansion and works well on the original non-Hermitian matrix, and hence it can be implemented more efficiently. Based on this idea, this paper considers three cases of the solution of linear systems and proposes a quantum linear system algorithm for general matrices, where *A* is not required to be sparse or square, which can effectively improve the computational efficiency and expand the application range of quantum computation. For the homogeneous linear equations, we design the corresponding quantum circuit to ensure the completeness of the solution, which supplies exponential speed-up over classical algorithms. Meanwhile, we modify the quantum phase estimation (QPE) circuit to determine the sign of the phase by setting a sign qubit, which can be widely applied to various quantum algorithms.

The rest of our paper is organized as follows. Section 2 analyzes a general model of classical identification system based on semi-tensor product and shows the detailed process of our quantum algorithms. In Section 3, we make a time complexity comparison between existing algorithms and our algorithms. Then, we perform a numerical simulation to clarify the process of quantum algorithm in Section 4. Finally, we conclude in Section 5.

## 2. Quantum Algorithms for System Identification

### 2.1. The Classical System Identification Problem

Consider a general discrete model of system identification as follows:(1)x(i+1)=Ax(i)+Bu(i)
where x(i) is an *n* dimensional system state of the *i*-th sampling, u(i) is the input with dimension *m*, *A* is an n×n system matrix and *B* is an n×m matrix. The goal of system identification is to estimate the matrices *A* and *B* from a set of inputs {u(i)} and states {x(i)}.

System identification problems can be expressed in terms of the semi-tensor product method [32]. As a kind of special matrix multiplication, the semi-tensor product generalizes the ordinary matrix multiplication to the general case. T⊗S denotes the Kronecker product of matrices $T_{m\times n}$ and $S_{p\times q}$, which is expressed as
(2)$$T⊗S=\left[\begin{array}{cccc} t_{1,1}S & t_{1,2}S & \cdots & t_{1,n}S\\ t_{2,1}S & t_{2,2}S & \cdots & t_{2,n}S\\ \vdots & \vdots & \vdots & \vdots \\ t_{m,1}S & t_{m,2}S & \cdots & t_{m,n}S \end{array}\right]$$

Just as a computational tool for solving the model, T⋉S denotes the semi-tensor product of matrices *T* and *S*:(3)$$T⋉S=\left(T⊗I_{l/n}\right)\left(S⊗I_{l/p}\right)$$
where l=lcm(n,p) is the least common multiple of *n* and *p*. The semi-tensor product is the generalization of matrix multiplication. When n=p, there are l=n=p and T⋉S=TS.

Define $V_{C}\left(S\right)=\left[\begin{array}{c} s_{1}\\ s_{2}\\ \vdots \\ s_{n} \end{array}\right]$, where $s_{i}$ is the *i*th column vector of the matrix *S*. Therefore, we may estimate *A* and *B* from a set of u(i) and x(i).
(4)$$\begin{array}{cc} x(i+1) & =Ax(i)+Bu(i)\\ \; & =\left[\begin{array}{c} a_{11}x_{1}\left(i\right)+\cdots +a_{1n}x_{n}\left(i\right)\\ \vdots \\ a_{n1}x_{1}\left(i\right)+\cdots +a_{nn}x_{n}\left(i\right) \end{array}\right]+\left[\begin{array}{c} b_{11}u_{1}\left(i\right)+\cdots +b_{1m}u_{m}\left(i\right)\\ \vdots \\ b_{n1}u_{1}\left(i\right)+\cdots +b_{nm}u_{m}\left(i\right) \end{array}\right]\\ \; & =x{\left(i\right)}^{T}⊗I_{n}\cdot V_{C}\left(A\right)+u{\left(i\right)}^{T}⊗I_{n}\cdot V_{C}\left(B\right)\\ \; & =x{\left(i\right)}^{T}⋉V_{C}\left(A\right)+u{\left(i\right)}^{T}⋉V_{C}\left(B\right)\\ \; & =\left(x{\left(i\right)}^{T},u{\left(i\right)}^{T}\right)⋉\left[\begin{array}{c} V_{C}\left(A\right)\\ V_{C}\left(B\right) \end{array}\right] \end{array}$$
where $x{\left(i\right)}^{T}$ is a 1×n matrix and $V_{C}\left(A\right)$ is an $n^{2}\times 1$ matrix. According to Equation (Equation 3), the least common multiple lcm$\left(n,n^{2}\right)=n^{2}$, and $x{\left(i\right)}^{T}⋉V_{C}\left(A\right)=\left(x{\left(i\right)}^{T}⊗I_{n}\right)\left(V_{C}\left(A\right)⊗I_{1}\right)$.

Suppose there are N+1 observed samples, and
(5)$$W=\left[\begin{array}{c} x(2)\\ \vdots \\ x(N+1) \end{array}\right],H=\left[\begin{array}{c} x{\left(1\right)}^{T},u{\left(1\right)}^{T}\\ \vdots \\ x{\left(N\right)}^{T},u{\left(N\right)}^{T} \end{array}\right],Y=\left[\begin{array}{c} V_{C}\left(A\right)\\ V_{C}\left(B\right) \end{array}\right].$$
Then we have
(6)$$\begin{array}{cc} H⋉ & Y=W\\ \; & ⇓\\ (H⊗ & I_{n})Y=W \end{array}$$
The Equation (Equation 6) is a linear system of equations, and the task is to find the solution *Y*. In the Equation (Equation 6), $H⊗I_{n}$ is an Nn×(m+n)n matrix, where *n* is the dimension of system states, *m* is the dimension of the input, and *N* is the number of samples. For the high-dimensional identification system, the time complexity of classical algorithms is enormous and existing quantum algorithms can not directly solve the non-square linear systems of equations. In order to reduce the cost of computing resources, it is necessary to propose a quantum algorithm for general linear equations.

### 2.2. The Quantum Linear System Algorithm for General Matrices

Inspired by the singular value estimation algorithm [23,28,29], we propose a quantum algorithm for general linear systems of equations as follows.

Given a general linear equation Ax=b, the singular value decomposition is
(7)$$A=\sum_{i}\sigma_{i}\mu_{i}\nu_{i}^{T}$$
where $A\in R^{m\times n}$, $x\in R^{n}$ and $b\in R^{m}$, $\sigma_{i}$ is the singular value of *A*, $\mu_{i}\in R^{m}$ and $\nu_{i}\in R^{n}$ are the left and right singular vectors, and $\mu_{i}^{T}\mu_{i}=\nu_{i}^{T}\nu_{i}=1,\mu_{i}^{T}\mu_{j}=\nu_{i}^{T}\nu_{j}=0\left(i\neq j\right)$.

Let the rank of *A* be *r*(r≤m,n) and the rank of [Ab] be *q*; the relation between the solution vector *x* of Ax=b and the r,q is: (8)approximatesolutionx^,r=nandr<q;uniquesolutionx,r=n=q;generalsolutionx¯,r<n.

The linear system of equations Ax=b can be solved by a mathematical optimization technique of minimizing the sum of squares of errors between the solution and the actual data, which is the so-called least squares method
(9)$$e={∥Ax-b∥}^{2}$$
In the Equation (Equation 7), $\left\{\mu_{i}\right\}\in R^{m}$ and $\left\{\nu_{i}\right\}\in R^{n}$ are a set of basis in *m* and *n* dimensional spaces. Therefore, *x* and *b* can be expressed as $x=\sum_{i}^{n}\alpha_{i}\nu_{i},b=\sum_{i}^{m}\beta_{i}\mu_{i}$, and
(10)$$\begin{array}{cc} e & ={∥Ax-b∥}^{2}\\ \; & ={∥\sum_{i=1}^{r}\sigma_{i}\mu_{i}\nu_{i}^{T}\sum_{i=1}^{n}\alpha_{i}\nu_{i}-\sum_{i=1}^{m}\beta_{i}\mu_{i}∥}^{2}\\ \; & ={∥\sum_{i=1}^{r}\left(\sigma_{i}\alpha_{i}-\beta_{i}\right)\mu_{i}-\sum_{i=r+1}^{m}\beta_{i}\mu_{i}∥}^{2}\\ \; & =\sum_{i=1}^{r}{\left(\sigma_{i}\alpha_{i}-\beta_{i}\right)}^{2}+\sum_{i=r+1}^{m}\beta_{i}^{2} \end{array}$$
When $\alpha_{i}=\beta_{i}/\sigma_{i}$, $e_{m}=\min{∥Ax-b∥}^{2}=\sum_{i=r+1}^{m}\beta_{i}^{2}$.

Note that when r<n, $\alpha_{i}\left(i=r+1,\ldots ,n\right)$ is not assigned, and the equation Ax=b has infinitely many solutions. In engineering, we usually want to find out the lowest energy solution state *x* with 〈x|x〉 minimality, that is
(11)$$\left\{\begin{array}{cc} \alpha_{i}=\beta_{i}/\sigma_{i}, & i\in [1,r]\\ \alpha_{i}=0, & i\in [r+1,n] \end{array}\right.$$

The goal is to convert the state $b=\sum_{i=1}^{m}\beta_{i}\mu_{i}$ to $x=\sum_{i=1}^{r}\beta_{i}/\sigma_{i}\nu_{i}$, whose detailed quantum process of our scheme is described as follows.

The following mappings to access to the data structure can be performed in O(polylog(mn)) time.
(12)$$\begin{array}{cc} \; & U_{P}:|\xi \rangle |0\rangle =\sum \xi_{i}|i\rangle |0\rangle \to \sum \xi_{i}|i,A_{i}\rangle \\ \; & U_{Q}:|0\rangle |\xi \rangle =\sum \xi_{j}|0\rangle |j\rangle \to \sum \xi_{j}|A_{F},j\rangle \end{array}$$
The data structure is based on an array of binary trees, each binary tree contains enough leaves that store the squared amplitudes of the corresponding matrix entry, which can be found in [28,33] with a detailed description of such a binary tree memory structure. In order to facilitate mathematical operation, we define two degenerate operators *P* and *Q* that operate only on valid input information |ξ〉, where the dimension of the input state |ξ〉|0〉 is reduced to the dimension of valid information |ξ〉, so *P* and *Q* are called degenerate operators. The maps *P* and *Q* append an arbitrary input state |ξ〉 to a register that encodes:(13)$$\begin{array}{cc} \; & P:|\xi \rangle =\sum \xi_{i}|i\rangle \to \sum \xi_{i}|i,A_{i}\rangle =|P\xi \rangle \\ \; & Q:|\xi \rangle =\sum \xi_{j}|j\rangle \to \sum \xi_{j}|A_{F},j\rangle =|Q\xi \rangle \end{array}$$
where $|i,A_{i}\rangle =\frac{1}{∥A_{i}∥}\sum_{j=1}^{n}A_{ij}|i,j\rangle$ and $|A_{F},j\rangle =\frac{1}{{∥A∥}_{F}}\sum_{i=1}^{m}∥A_{i}∥|i,j\rangle$. That is, $P=\sum_{i=1}^{m}\left|i\rangle \right|A_{i}\rangle \langle i|$ and $Q=\sum_{j=1}^{n}|A_{F}\rangle |j\rangle \langle j|$.

Based on the above definition, it is easy to obtain ${\left(P^{†}Q\right)}_{ij}=\langle i,A_{i}|A_{F},j\rangle =\frac{A_{ij}}{∥A_{F}∥}$. Similarly, it follows that *P* and *Q* have orthonormal columns and thus $P^{†}P=I_{m}$ and $Q^{†}Q=I_{n}$. Let $S=\left(2PP^{†}-I\right)\left(2QQ^{†}-I\right)$, when m=n, we can obtain
(14)$$\begin{array}{cc} \; & SQ|\nu_{i}\rangle =\frac{2\sigma_{i}}{{∥A∥}_{F}}P|\mu_{i}\rangle -Q|\nu_{i}\rangle \\ \; & SP|\mu_{i}\rangle =\left(\frac{4\sigma_{i}^{2}}{{∥A∥}_{F}^{2}}-1\right)P|\mu_{i}\rangle -\frac{2\sigma_{i}}{{∥A∥}_{F}}Q|\nu_{i}\rangle \end{array}$$
The eigenvalues of *S* are $e^{\pm 2\pi i\varphi_{i}}$, and the corresponding eigenvectors are $\omega_{i}^{\pm }|w_{i}^{\pm }\rangle =-P|\mu_{i}\rangle +e^{\mp 2\pi i\varphi_{i}}Q|\nu_{i}\rangle$, where $\varphi_{i}$ is the phase of eigenvalues and $\omega_{i}^{\pm }$ is the norm of eigenvectors. Then, it can be obtained
(15)$$\begin{array}{cc} \; & Q|\nu_{i}\rangle =\frac{1}{2i\sin(\pi \varphi_{i})}(\omega_{i}^{+}|w_{i}^{+}\rangle -\omega_{i}^{-}|w_{i}^{-}\rangle )\\ \; & P|\mu_{i}\rangle =\frac{1}{2i\sin(\pi \varphi_{i})}(e^{\pi i\varphi_{i}}\omega_{i}^{+}|w_{i}^{+}\rangle -e^{-\pi i\varphi_{i}}\omega_{i}^{-}|w_{i}^{-}\rangle ) \end{array}$$
Through phase rotation, the process of $\sum_{i=1}^{n}\beta_{i}|\mu_{i}\rangle \mapsto \sum_{i=1}^{n}\beta_{i}|\nu_{i}\rangle$ is achievable [28,29]. It is worth noting that the above step is avoidable for the case of the coefficient matrix *A* being Hermitian. At this point, the singular value decomposition is $A=\sum_{i}\sigma_{i}\mu_{i}\mu_{i}^{T}$, and the task is to convert $|b\rangle =\sum_{i}\beta_{i}|\mu_{i}\rangle$ to the solution $|x\rangle =\sum_{i}\beta_{i}/\sigma_{i}|\mu_{i}\rangle$ such that Ax=b. However, for the case of non-Hermitian, it is necessary to realize the transformation of quantum states $|\mu_{i}\rangle$ to $|\nu_{i}\rangle$.

For a general linear system of equations with m≠n, the above derivation will have some changes. For i∈[1,r], Equations (14) and (15) are valid. While i>r, $A|\nu_{i}\rangle =0,A^{†}|\mu_{i}\rangle =0$, and we can obtain
(16)$$\begin{array}{cc} SQ|\nu_{i}\rangle & =\left(2PP^{†}-I\right)\left(2QQ^{†}-I\right)Q|\nu_{i}\rangle \\ \; & =\left(2PP^{†}-I\right)Q|\nu_{i}\rangle \\ \; & =\frac{2}{{∥A∥}_{F}}PA|\nu_{i}\rangle -Q|\nu_{i}\rangle \\ \; & =-Q|\nu_{i}\rangle , \end{array}$$
and
(17)$$\begin{array}{cc} SP|\mu_{i}\rangle & =\left(2PP^{†}-I\right)\left(2QQ^{†}-I\right)P|\mu_{i}\rangle \\ \; & =\left(2PP^{†}-I\right)(\frac{2}{{∥A∥}_{F}}QA^{†}|\mu_{i}\rangle -P|\mu_{i}\rangle )\\ \; & =\left(2PP^{†}-I\right)(-P|\mu_{i}\rangle )\\ \; & =-P|\mu_{i}\rangle . \end{array}$$
At this point, $e^{2\pi i\varphi_{i}}=-1$ is the eigenvalue of *S*, that is, $\varphi_{i}=\pm 1/2$, and the corresponding eigenvectors are $Q|\nu_{i}\rangle$ and $P|\mu_{i}\rangle$. In order to achieve $|b\rangle =\sum_{i=1}^{m}\beta_{i}|\mu_{i}\rangle =\sum_{i=1}^{r}\beta_{i}|\mu_{i}\rangle +\sum_{i=r+1}^{m}\beta_{i}|\mu_{i}\rangle \mapsto |x\rangle =\sum_{i=1}^{r}\beta_{i}/\sigma_{i}|\nu_{i}\rangle$, we first need to eliminate the formula $\sum_{i=r+1}^{m}\beta_{i}|\mu_{i}\rangle$.

Based on these definitions, we show the basic procedure of our algorithm:Preparing the initial quantum state $|b\rangle =\Sigma_{i=1}^{m}b_{i}|i\rangle$, which can be represented as:
(18)$$|b\rangle =\sum_{i=1}^{m}\beta_{i}|\mu_{i}\rangle$$Apply *P* in the initial state |b〉
(19)$$\begin{array}{cc} P|b\rangle & =\sum_{i=1}^{m}\beta_{i}P|\mu_{i}\rangle \\ \; & =\sum_{i=1}^{r}\frac{\beta_{i}}{2i\sin(\pi \varphi_{i})}(e^{\pi i\varphi_{i}}\omega_{i}^{+}|w_{i}^{+}\rangle -e^{-\pi i\varphi_{i}}\omega_{i}^{-}|w_{i}^{-}\rangle )+\sum_{i=r+1}^{m}\beta_{i}P|\mu_{i}\rangle \end{array}$$Perform phase estimation on input P|b〉 for $S=\left(2PP^{†}-I\right)\left(2QQ^{†}-I\right)$, as shown in Figure 1, then we obtain the following state
(20)$$\begin{array}{c} \sum_{i=1}^{r}\frac{\beta_{i}}{2i\sin(\pi \varphi_{i})}(e^{\pi i\varphi_{i}}\omega_{i}^{+}|w_{i}^{+},\varphi_{i}\rangle -e^{-\pi i\varphi_{i}}\omega_{i}^{-}|w_{i}^{-},-\varphi_{i}\rangle )+\sum_{i=r+1}^{m}\beta_{i}P|\mu_{i}\rangle |\pm \frac{1}{2}\rangle , \end{array}$$
where $e^{2\pi i\varphi_{i}}$ is the eigenvalue of *S* and $|\frac{1}{2}\rangle =|01000\ldots \rangle ,|-\frac{1}{2}\rangle =|11000\ldots \rangle$.Apply a phase shift operator controlled by the phase $\varphi_{i}$, then we obtain
(21)$$\sum_{i=1}^{r}\frac{\beta_{i}}{2i\sin(\pi \varphi_{i})}(\omega_{i}^{+}|w_{i}^{+},\varphi_{i}\rangle -\omega_{i}^{-}|w_{i}^{-},-\varphi_{i}\rangle )+\sum_{i=r+1}^{m}\beta_{i}P|\mu_{i}\rangle |\pm \frac{1}{2}\rangle$$Perform a controlled rotation on the ancillary qubit based on the register storing phase value $\varphi_{i}$ and will obtain
(22)$$\sum_{i=1}^{r}\frac{\beta_{i}}{2i\sin(\pi \varphi_{i})}(\omega_{i}^{+}|w_{i}^{+},\varphi_{i}\rangle -\omega_{i}^{-}|w_{i}^{-},-\varphi_{i}\rangle )\left(\frac{t}{\sigma_{i}}|0\rangle +\sqrt{1-\frac{t^{2}}{\sigma_{i}^{2}}}|1\rangle \right)+\sum_{i=r+1}^{m}\beta_{i}P|\mu_{i}\rangle |\pm \frac{1}{2}\rangle |1\rangle ,$$
where $\sigma_{i}=\cos\left(\pi \varphi_{i}\right){∥A∥}_{F}$ and $t=min_{i}\left|\sigma_{i}\right|,i\in \left[1,r\right]$.Apply the inverse transformation of step 3 to obtain
(23)$$\begin{array}{cc} \; & \sum_{i=1}^{r}\frac{\beta_{i}}{2i\sin(\pi \varphi_{i})}(\omega_{i}^{+}|w_{i}^{+}\rangle -\omega_{i}^{-}|w_{i}^{-}\rangle )\left(\frac{t}{\sigma_{i}}|0\rangle +\sqrt{1-\frac{t^{2}}{\sigma_{i}^{2}}}|1\rangle \right)+\sum_{i=r+1}^{m}\beta_{i}P|\mu_{i}\rangle |\rangle |1\rangle \\ \; & =\sum_{i=1}^{r}\beta_{i}Q|\nu_{i}\rangle \left(\frac{t}{\sigma_{i}}|0\rangle +\sqrt{1-\frac{t^{2}}{\sigma_{i}^{2}}}|1\rangle \right)+\sum_{i=r+1}^{m}\beta_{i}P|\mu_{i}\rangle |1\rangle \end{array}$$Measure the ancillary register. When the measurement result is |0〉, the quantum state will collapse to
(24)$$\sum_{i=1}^{r}\beta_{i}/\sigma_{i}Q|\nu_{i}\rangle$$Apply the inverse of *Q* and we will obtain the desired state
(25)$$\sum_{i=1}^{r}\beta_{i}/\sigma_{i}|\nu_{i}\rangle ,$$
which is the particular solution of the equation Ax=b, that is, the lowest energy solution state.

The quantum gate circuit of our quantum algorithm is shown in Figure 2.

Note that the actual phase value is $\varphi_{i}\in \left(-1,1\right)$, which serves as the control qubits of the phase shift operation in the step 4, while the previous quantum phase estimation algorithm outputs phase value $\varphi_{i}\in \left(0,1\right)$. Therefore, we design a modified quantum phase estimation circuit to determine the sign of the phase in Figure 1. In the modified QPE circuit, we can estimate the phase value in the range φ∈(−1,1).

For the case of φ∈[0,1), $\varphi =\sum_{j=1}^{k}2^{-j+1}x_{j}$. Since 0≤φ<1, it is easy to obtain $x_{1}=0$.

While φ∈(−1,0), we can obtain $e^{2\pi i2^{j}\varphi }=e^{2\pi i2^{j}\left(2+\varphi \right)}$, where j=−1,…,k−1. Let ϕ=2+φ∈(1,2), the modified QSV circuit outputs $\phi =\sum_{j=1}^{k}2^{-j+1}x_{j}$. It is known that ϕ∈(1,2), thus we obtain $x_{1}=1$ and $\varphi =\sum_{j=1}^{k}2^{-j+1}x_{j}-2=\sum_{j=2}^{k}2^{-j+1}x_{j}-1$.

Therefore, we can obtain
(26)$$\varphi =\left\{\begin{array}{cc} \sum_{j=1}^{k}2^{-j+1}x_{j}\;, & x_{1}=0\\ \sum_{j=2}^{k}2^{-j+1}x_{j}-1\;, & x_{1}=1 \end{array}\right.$$
that is, $\varphi =\sum_{j=2}^{k}2^{-j+1}x_{j}-x_{1}$.

### 2.3. The Quantum Algorithm for Homogeneous Linear Equations

For the condition of r<n, the equation Ax=b has infinitely many solutions. Therefore, in order to obtain general solutions of the equation Ax=b, we need to solve the homogeneous linear equation Ax=0.

Since $A=\sum_{i=1}^{n}\sigma_{i}\mu_{i}\nu_{i}^{T}$ and $x=\sum_{i=1}^{n}\alpha_{i}\nu_{i}$, we can obtain $Ax=\sum_{i=1}^{n}\sigma_{i}\alpha_{i}\mu_{i}$. Let $\nu_{i}\left(i\in \left[r+1,n\right]\right)$ be the right singular vector corresponding to $\sigma_{i}=0$, when $\sigma_{i}=0$ or $\alpha_{i}=0$, Ax=0 is valid, that is, $x=\sum_{i=r+1}^{n}\alpha_{i}\nu_{i}$. The description of solving homogeneous linear equations is essentially just finding the projection of a state onto the ground state for an operator [34]. Through the quantum circuit shown in Figure 3, we obtain the combination of the eigenvectors corresponding to $\sigma_{i}=0$ and make the solution of homogeneous linear equations complete.

Based on QSVE, our quantum algorithms are sparsity-independent and may be applied to non-square dense matrices.

## 3. Algorithms Complexity Analysis

Then, we analyse the time complexity of our quantum algorithms.

The time complexity of our scheme includes the following two parts: quantum data generation and the quantum algorithm process. On the one hand, relying on a binary tree memory structure detailed as described in [28,33], where the matrix entries associated with matrix $A_{m\times n}$ are stored as suitable data structure, the oracle from classical data to quantum data can be implemented efficiently in time $O(\log^{2}mn)$ and the data structure size is O(wlogmn) where *w* is the number of non zero entries in *A*. On the other hand, based on the quantum singular value estimation algorithm, our algorithm achieves a runtime O(κ polylog(mn)/δ), where κ is the condition number of the coefficient matrix *A*, and δ denotes the precision parameter.

Define that the additive error achieved in output state $\tilde{x}$ is ϵ, which means if *x* is the exact result and $\tilde{x}$ is the result obtained from quantum algorithms, then $∥x-\tilde{x}∥\leq \epsilon$. In order to achieve accuracy ϵ, the precision parameter of our algorithm needs to reach $\delta =\epsilon /(\kappa ∥A{∥}_{F})$. Assuming the spectral norm ${∥A∥}_{*}$ is bounded by a constant, since ${∥A∥}_{F}\leq \sqrt{r}{∥A∥}_{*}$, we have ${∥A∥}_{F}=O\left(\sqrt{r}\right)$, where *r* is the rank of matrix *A*. Therefore, our quantum algorithm has the time complexity of $O(\kappa^{2}\sqrt{r}$polylog(mn)/ϵ). Remarkably, when *A* is sparse, exponential acceleration is achievable.

In the quantum algorithm of homogeneous linear equations, the time complexity of QSVE is O( polylog(mn)/ϵ). In view of the success probability of the ancillary register collapses to |1〉, we need to repeat the coherent computation n/(n−r) times on average. Therefore, the runtime of the quantum homogeneous linear equation algorithm is given by O(n polylog(mn)/((n−r)ϵ)).

After obtaining the quantum state |x〉 corresponding to the solution of Ax=b, we need to simulate the subsequent system states through the identified system, where $x=\left[\begin{array}{c} V_{C}\left(A\right)\\ V_{C}\left(B\right) \end{array}\right]$. According to Equations (Equation 4)–(6), we can obtain
(27)x(i+1)=〈ζ(i)|x〉
where $\zeta \left(i\right)=\left[x{\left(i\right)}^{T},u{\left(i\right)}^{T}\right]⊗I_{n}$. The inner product between pairs of states can be implemented in time O(ploylog(m+n)) by the swap text algorithm [35]. Therefore, we can predict the system state at the next moment based on the known system state x(i) and input u(i).

For the case of general matrix $A_{m\times n}$, previous quantum algorithms generally convert *A* to a Hermitian matrix:(28)$$H=\left[\begin{array}{cc} 0 & A\\ A^{†} & 0 \end{array}\right]$$
Based on QSVE, the quantum algorithm of $H\left[\begin{array}{c} 0\\ x \end{array}\right]=\left[\begin{array}{c} b\\ 0 \end{array}\right]$ has the time complexity of $O(\kappa^{2}∥H{∥}_{F}$ poly$\log{\left(m+n\right)}^{2}/\epsilon )$. In addition, the time complexity of our scheme is $O(\kappa^{2}∥A{∥}_{F}$ polylog(mn)/ϵ), where ${∥H∥}_{F}=\sqrt{2}{∥A∥}_{F}$. Let the runtime of our scheme be *T*, so the runtime of existing SVE-based quantum algorithms is $T^{\prime }=\sqrt{2}\frac{\mathrm{poly}\log{\left(m+n\right)}^{2}}{\mathrm{poly}\log(mn)}T$. For the large-scale linear system of equations, there is $T^{\prime }\approx \sqrt{2}T$. Compared to existing quantum algorithms, our scheme can reduce the time complexity of the linear system of equations with a non-square dense matrix.

## 4. Numerical Simulation

To clarify the process of our algorithm and prove the feasibility of algorithms, we perform simulation on an illustrative example.

For simplicity, we consider a first-order discrete model of classical system as follows:(29)x(i+1)=ax(i)+du(i)
where x(i) is the system state of the *i*-th sampling and u(i) is the input state. The goal of system identification is to estimate coefficients *a* and *d* from a set of u(i) and x(i). Assuming that the initial system state x(1)=3, the input states u={4,3,0} and the evolved system states x(2)=−4,x(3)=x(4)=0, the mathematical model can be transformed into a general linear systems of equations Ax=b, where $A=\left[\begin{array}{cc} 3 & 4\\ -4 & 3\\ 0 & 0 \end{array}\right],b=\left[\begin{array}{c} -4\\ 0\\ 0 \end{array}\right]$. The maps $U_{P}$ and $U_{Q}$ append an arbitrary input state to a register that encodes the Equation (Equation 12), which can be realized by quantum gate circuits in Figure 4, and matrix forms of these maps are $P=\frac{1}{5}\left[\begin{array}{ccc} 3 & 0 & 0\\ 4 & 0 & 0\\ 0 & -4 & 0\\ 0 & 3 & 0\\ 0 & 0 & \frac{5\sqrt{2}}{2}\\ 0 & 0 & \frac{5\sqrt{2}}{2} \end{array}\right],Q=\frac{\sqrt{2}}{2}\left[\begin{array}{cc} 1 & 0\\ 0 & 1\\ 1 & 0\\ 0 & 1\\ 0 & 0\\ 0 & 0 \end{array}\right]$.

The following shows the detailed procedure of the numerical simulation:Preparing the initial state |b〉=|0〉.Apply *P* in the initial state |b〉, $P|b\rangle =\frac{3}{5}|0\rangle +\frac{4}{5}|1\rangle$.Perform phase estimation on P|b〉 for $S=\left(2PP^{†}-I\right)\left(2QQ^{†}-I\right)=\frac{1}{25}\left[\begin{array}{cccccc} 0 & 0 & -7 & 24 & 0 & 0\\ 0 & 0 & 24 & 7 & 0 & 0\\ 7 & -24 & 0 & 0 & 0 & 0\\ -24 & -7 & 0 & 0 & 0 & 0\\ 0 & 0 & 0 & 0 & 0 & 25\\ 0 & 0 & 0 & 0 & 25 & 0 \end{array}\right]$, then we obtain the following state
(30)$$\begin{array}{c} -\frac{3\sqrt{2}i}{10}\left(e^{\frac{1}{4}\pi i}\omega_{1}|w_{1},\frac{1}{4}\rangle -e^{-\frac{1}{4}\pi i}\omega_{2}|w_{2},-\frac{1}{4}\rangle \right)-\frac{2\sqrt{2}i}{5}\left(e^{\frac{3}{4}\pi i}\omega_{3}|w_{3},\frac{3}{4}\rangle -e^{-\frac{1}{4}\pi i}\omega_{4}|w_{4},-\frac{3}{4}\rangle \right), \end{array}$$
where the eigenvalues of *S* are $\lambda_{i}=i,i,-i,-i,1,-1$, and $\omega_{i}|w_{i}\rangle$ is the corresponding eigenvector.Change the phase, then we obtain
(31)$$-\frac{3\sqrt{2}i}{10}\left(\omega_{1}|w_{1},\frac{1}{4}\rangle -\omega_{2}|w_{2},-\frac{1}{4}\rangle \right)-\frac{2\sqrt{2}i}{5}\left(\omega_{3}|w_{3},\frac{3}{4}\rangle -\omega_{4}|w_{4},-\frac{3}{4}\rangle \right)$$Perform a controlled rotation on the ancillary qubit based on the register storing phase value:
(32)$$-\frac{3\sqrt{2}i}{10}\left(\omega_{1}|w_{1},\frac{1}{4}\rangle -\omega_{2}|w_{2},-\frac{1}{4}\rangle \right)\left(\frac{1}{5}|0\rangle +\frac{2\sqrt{6}}{5}|1\rangle \right)-\frac{2\sqrt{2}i}{5}\left(\omega_{3}|w_{3},\frac{3}{4}\rangle -\omega_{4}|w_{4},-\frac{3}{4}\rangle \right)\left(-\frac{1}{5}|0\rangle +\frac{2\sqrt{6}}{5}|1\rangle \right)$$
where $\sigma_{1}=\cos\left(\pm \frac{1}{4}\pi \right){∥A∥}_{F}=5,\sigma_{2}=\cos\left(\pm \frac{3}{4}\pi \right){∥A∥}_{F}=-5$.Apply the inverse transformation of step 3 to obtain
(33)$$\begin{array}{c} -\frac{3\sqrt{2}i}{10}(\omega_{1}|w_{1}\rangle -\omega_{2}|w_{2}\rangle )\left(\frac{1}{5}|0\rangle +\frac{2\sqrt{6}}{5}|1\rangle \right)-\frac{2\sqrt{2}i}{5}(\omega_{3}|w_{3}\rangle -\omega_{4}|w_{4}\rangle )\left(-\frac{1}{5}|0\rangle +\frac{2\sqrt{6}}{5}|1\rangle \right) \end{array}$$Apply the inverse of *Q* and we will obtain the desired state
(34)$$\frac{3}{5}|0\rangle \left(\frac{1}{5}|0\rangle +\frac{2\sqrt{6}}{5}|1\rangle \right)-\frac{4}{5}|1\rangle \left(-\frac{1}{5}|0\rangle +\frac{2\sqrt{6}}{5}|1\rangle \right)$$Measure the ancillary register. When the result is |0〉, the quantum state will collapse to
(35)$$\frac{3}{5}|0\rangle +\frac{4}{5}|1\rangle$$
that is proportional to the solution of the equation Ax=b, so we obtain $a=\frac{3}{5}C$ and $d=\frac{4}{5}C$. Substituting *a* and *d* into Equation (Equation 29), we obtain $C=-\frac{4}{5}$. So far, the first-order discrete identification model is:
(36)$$x\left(i+1\right)=-\frac{12}{25}x\left(i\right)-\frac{16}{25}u\left(i\right)$$

We simulated a 6-qubit quantum circuit diagram on the Origin Cloud, as shown in Figure 5.

According to the simulation result in Figure 6, when the ancillary qubit q[5] = 0 and the register storing phase information is restored to q[0] = q[1] = q[2] = 0, the probabilities of the output qubit q[4] are P{|0〉}=0.086 and P{|1〉}=0.16. Therefore, the amplitudes of q[4] are $A\{|0\rangle \}=\sqrt{P\{|0\rangle \}/(P\{|0\rangle \}+P\{|1\rangle \})}=0.59$ and $A\{|1\rangle \}=\sqrt{P\{|1\rangle \}/(P\{|0\rangle \}+P\{|1\rangle \})}=0.81$, and the solution quantum state is q[4]=0.59|0〉+0.81|1〉, which is consistent with the expected quantum state based on our algorithm.

As a comparison, we simulated a second-order discrete system identification model $x\left(i+1\right)=A^{\prime }x\left(i\right)+B^{\prime }u\left(i\right)$, where $A^{\prime }$ is a 2×2 matrix and $B^{\prime }$ is a 2×1 matrix. Assuming that the initial system state $x\left(1\right)=\left[\begin{array}{c} 1\\ 0 \end{array}\right]$, the input u={1,2,3} and the evolved system states $x\left(2\right)=\left[\begin{array}{c} 1\\ 1 \end{array}\right]$, the mathematical model can be transformed into Ax=b, where $A=\left[\begin{array}{cccccc} 1 & 0 & 0 & 0 & 1 & 0\\ 0 & 1 & 0 & 0 & 0 & 1 \end{array}\right]$ and $b=\left[\begin{array}{c} 1\\ 1 \end{array}\right]$. Due to insufficient samples, the linear equation has infinite solutions. Consider the structural features of the coefficient matrix *A*, which is reduced to $\left[\begin{array}{cccc} 1 & 0 & 1 & 0\\ 0 & 1 & 0 & 1 \end{array}\right]$.

When the ancillary qubit q[6] = 0 and the register q[5] is restored to 0, the probabilities of the output qubits q[4] and q[3] are shown in the Figure 7. Thus, the solution quantum state is |x〉=0.493|00〉+0.5|01〉+0.516|10〉+0.485|11〉, which is the lowest energy solution among infinitely many solutions. So far, according to the existing samples, the second-order discrete identification model with the lowest energy is: $x\left(i+1\right)=\left[\begin{array}{cc} 0.493 & 0\\ 0.5 & 0 \end{array}\right]x\left(i\right)+\left[\begin{array}{c} 0.516\\ 0.485 \end{array}\right]u\left(i\right)$.

## 5. Conclusions

This paper develops a quantum algorithm of general linear equations for solving classical system identification problems. Our scheme can be finished in time $O(\kappa^{2}\sqrt{r}$polylog(mn)/ϵ) for an m×n dimensional linear systems of equations Ax=b, where κ is the condition number of the linear equation, *r* is the rank of the matrix *A* and ϵ is the precision parameter, which is superior to existing algorithms. For the linear equation with non-square coefficient matrix, we discuss three cases of solutions, including the unique solution, approximate solution and infinitely many solutions. Our algorithm can obtain the unique solution, the approximate solution with the minimum error and the lowest energy solution among infinitely many solutions, which adapts to all cases of linear systems of equations. For the case of infinitely many solutions, we design a quantum circuit to obtain general solutions in time O(n polylog(mn)/((n−r)ϵ)), which can achieve an exponential improvement over classical algorithms. In addition, we design a modified QPE circuit to obtain a wider range of phase values, which can expand the application range of quantum phase estimation.

Based on QSVE, our algorithms is sparsity-independent compared with HHL algorithm. Meanwhile, we have extended the existing quantum linear system algorithms to general equations, which can effectively enrich the application area of linear systems of equations. For large-scale linear systems, such as machine learning, numerical calculation of partial differential equations, etc., our algorithms will have a wider range of applications and is of research significance. In the future work, we will focus on how our algorithms are implemented on quantum computers and how to apply on them to real practical problems.

## Figures and Tables

**Figure 1 entropy-24-00893-f001:**
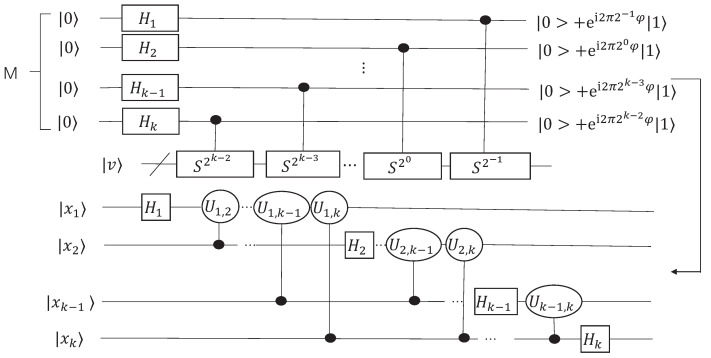
The modified quantum circuit for phase estimation. Set the $|x_{1}\rangle$ to be the sign bit, $|x_{1}\rangle =|0\rangle$ means $\varphi_{i}$ is a positive value, otherwise it is negative. The state $|\varphi_{i}\rangle =|x_{1}\rangle_{1}|x_{2}\rangle_{2}\ldots {|x_{k}\rangle }_{k}$ and the phase value $\varphi_{i}=\sum_{j=2}^{k}2^{-j+1}x_{j}-x_{1}$. The quantum circuit can estimate the phase value $\varphi_{i}\in \left(-1,1\right)$.

**Figure 2 entropy-24-00893-f002:**
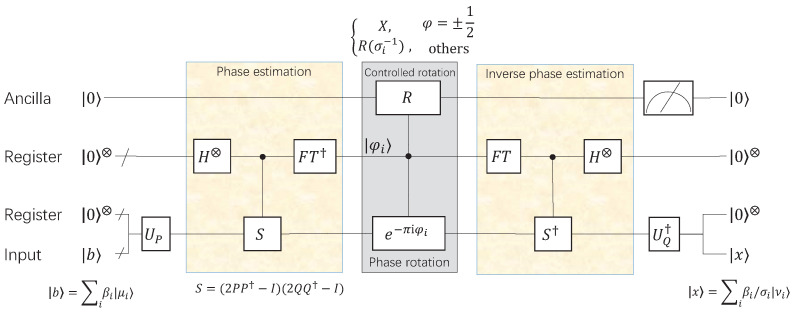
The quantum gate circuit of the particular solution of the equation Ax=b. The operator *R* is a quantum controlled rotation gate. When the phase $\varphi_{i}=\pm \frac{1}{2}$, *R* is a NOT gate, otherwise $R=R\left(\sigma_{i}^{-1}\right)=\left[\begin{array}{cc} 1/\sigma_{i} & \sqrt{1-1/\sigma_{i}^{2}}\\ \sqrt{1-1/\sigma_{i}^{2}} & -1/\sigma_{i} \end{array}\right]$, where $\sigma_{i}=\cos\left(\pi \varphi_{i}\right){∥A∥}_{F}$.

**Figure 3 entropy-24-00893-f003:**
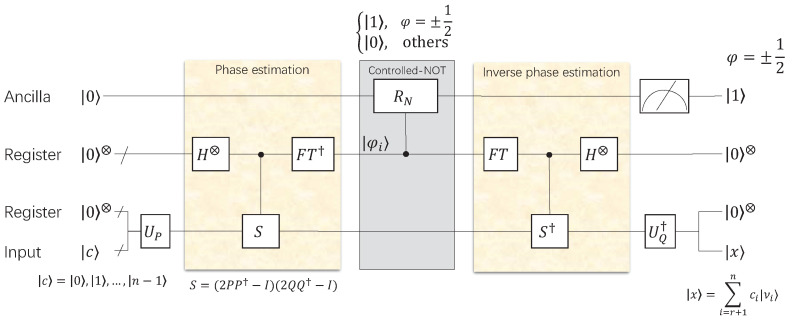
The quantum gate circuit of homogeneous linear equation Ax=0. The input state is $|c\rangle =\sum_{i=1}^{n}c_{i}|\nu_{i}\rangle$ and the output state is $|x\rangle =\sum_{i=r+1}^{n}c_{i}|\nu_{i}\rangle$, where |x〉 is the combination of right singular vectors corresponding to the singular value 0 of *A* contained in |c〉. When the phase $\varphi_{i}=\pm 1/2$, the output of the controlled-NOT gate $R_{N}$ is |1〉, otherwise it outputs |0〉. We can obtain the solution of the Ax=0 when an arbitrary input |c〉 contains right singular vectors of *A*. In order to ensure that the output |x〉 is complete, we input *n* linearly independent |c〉=|0〉,|1〉…|n〉. In addition, the arbitrary *r* linearly independent $x_{i}$ can form the solution vector basis of the homogeneous linear equation.

**Figure 4 entropy-24-00893-f004:**
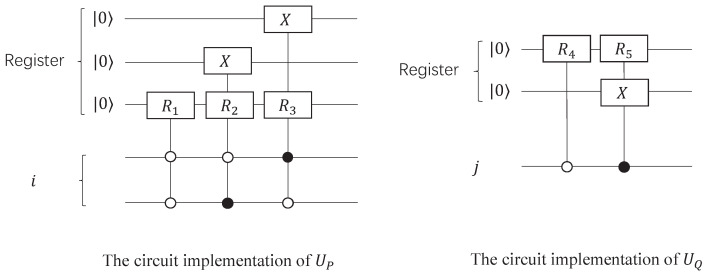
Quantum circuits implementation of $U_{P}$ and $U_{Q}$. The maps $U_{P}$ and $U_{Q}$ consist of quantum control gates, where $R_{1}=\frac{1}{5}\left[\begin{array}{cc} 3 & 4\\ 4 & -3 \end{array}\right],R_{2}=\frac{1}{5}\left[\begin{array}{cc} -4 & 3\\ 3 & 4 \end{array}\right]$, $R_{3}=R_{4}=R_{5}=H=\frac{\sqrt{2}}{2}\left[\begin{array}{cc} 1 & 1\\ 1 & -1 \end{array}\right]$ and the $X=\left[\begin{array}{cc} 0 & 1\\ 1 & 0 \end{array}\right]$ is the quantum inverse gate.

**Figure 5 entropy-24-00893-f005:**
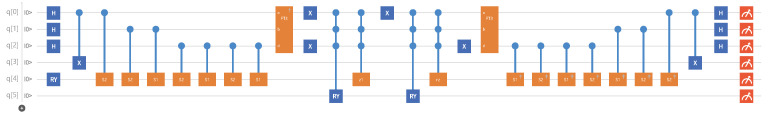
The 6-qubit quantum circuit diagram on the Origin Cloud. The quantum circuit is based on quantum phase estimation, where q[5] is the ancillary qubit and q[4] is the register storing input and output information. From left to right, the controlled rotation gate is $RY_{1}=\frac{1}{5}\left[\begin{array}{cc} 3 & -4\\ 4 & 3 \end{array}\right]$ that generates the initial state P|b〉, $RY_{2}=\frac{1}{\sigma_{1}}\left[\begin{array}{cc} t & -\sqrt{\sigma_{1}^{2}-t^{2}}\\ \sqrt{\sigma_{1}^{2}-t^{2}} & t \end{array}\right]$ and $RY_{3}=\frac{1}{\sigma_{2}}\left[\begin{array}{cc} t & -\sqrt{\sigma_{2}^{2}-t^{2}}\\ \sqrt{\sigma_{2}^{2}-t^{2}} & t \end{array}\right]$, where $t=\frac{5\sqrt{2}}{2}$. The controlled gates $s_{1}=\frac{1}{25}\left[\begin{array}{cc} -7 & 24\\ 24 & 7 \end{array}\right],s_{2}=\frac{1}{25}\left[\begin{array}{cc} 7 & -24\\ -24 & -7 \end{array}\right],z_{1}=\left[\begin{array}{cc} e^{-\frac{1}{4}\pi i} & 0\\ 0 & e^{-\frac{1}{4}\pi i} \end{array}\right]$ and $rz=\left[\begin{array}{cc} e^{-\frac{3}{4}\pi i} & 0\\ 0 & e^{-\frac{3}{4}\pi i} \end{array}\right]$ are applied on the register q[4].

**Figure 6 entropy-24-00893-f006:**
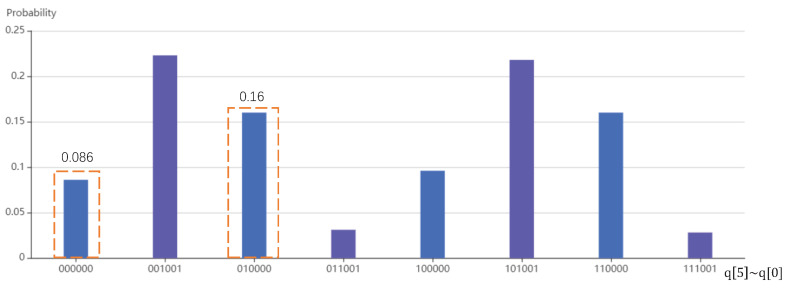
The simulation result of the quantum circuit.

**Figure 7 entropy-24-00893-f007:**
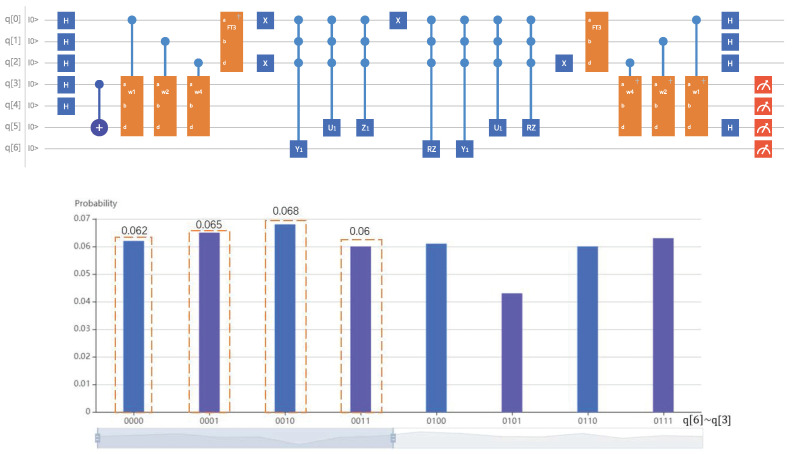
Quantum circuit and simulation result of the two-dimensional discrete system identification model.

## Data Availability

Data are contained within the article.

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
