# Peer review of "Quantum Linear System Algorithm for General Matrices in System Identification"

_entropy, 2022, doi:10.3390/e24070893_

Round 1

Reviewer 1 Report

In the abstract, the authors’ algorithm has complexity $\tilde{O}(\kappa^2 \sqrt{r}/\epsilon) for general $m\times n$ linear systems. They claimed that this achieves a polynomial improvement compared with existing quantum algorithms. I cannot agree with this. The current best quantum algorithm has a complexity linear in $\kappa$ and $\|A\|_F$, the dependence on $\epsilon$ is polylog. So this paper’s algorithm is worse. On page 2, it is claimed that “Existing quantum algorithms cannot directly be adapted to general matrices, …” I have to say that this is not correct. The current quantum algorithms work for all matrices. Please check this paper https://arxiv.org/abs/1804.01973. This paper uses the technique of SVE, this idea has already been considered in https://arxiv.org/abs/1704.06174.

Reviewer 2 Report

The paper presents a quantum algorithm for solving linear systems of  equations with general matrices. Next to presenting and analysing the algorithm, the authors present a possible application for the suggested algorithm, namely, system identification.

My major concern with the paper in its current form is that some steps in the derivation are only vaguely described which makes it impossible for me to verify whether the algorithm really works as claimed. I kindly ask the authors to revise their manuscript taking into account the following issues:

1. Figure 2 implements P and Q^\dagger as operators in the quantum circuit, which means that they must be unitary. However, P and Q are not defined explicitly throughout the paper. Section 2.2 describes them as "degenerate operators" (please define!) and defines their action in Eq. (12). In the numerical example given in Section 4, P and Q are rectangular matrices with a sketch of their implementation given in Figure 4. Please elaborate on the definition of P and Q and their implementation for the general case, i.e., not just give the circuits from Figure 4 but also explain how it is derived from matrix A and vector b in general. Also adjust Figure 2 to prevent any confusion due to P and Q^\dagger not being unitary operators. Concerning the concrete example given in Figure 4.

2. Another issue is the description of HHL in the introduction and the complexity analysis of the proposed algorithm. The authors state that HHL "find(s) the solution vector x such that Ax=b". In fact, the crux of HHL is that reading-out the solution vector x is inefficient but, instead, extracting only a scalar derived quantity x^\dagger M x with M being a sparse operator can be done efficiently. Please adjust this statement in the introduction. This issue is connected with a concern about the authors' algorithm. The presented numerical example has two parameters, 'a' and 'd', that need to be determined via read-out from the quantum experiment. The complexity analysis in Section 3 does not provide enough information about the complexity of reading out the full solution to the problem (matrices A and B) and also leaves it unclear how the action of matrices P and Q is realised efficiently for arbitrary matrices (see also my previous comment on elaborating on the implementation of P and Q). In my understanding, step P|0> can be seen as the state preparation step in HHL and other quantum linear solver algorithms which is, for general vectors 'b', up to exponentially expensive. Please elaborate on this issue.

3. The derivation from Eq. (4) to (6) needs more explanation.

4. I encourage the authors to add one more example in which the parameters 'a' and 'd' are matrices instead of scalar values. This should be doable with today's quantum computer simulators which can simulate quantum algorithms with more than 30 qubits efficiently. If possible, it would also be interesting to see the performance of the quantum algorithm given in Figure 5 on a real quantum computer.

Round 2

Reviewer 1 Report

I am happy with the response.

Author Response

The typos and mistakes have also been double-checked and the whole manuscript has been revised carefully. We would like to express our gratitude to you for your excellent work and constructive suggestions to improve the quality of the manuscript.

Reviewer 2 Report

Dear authors,

thank you very much for taking into account all my comments in the revised version of the manuscript which I consider appropriate for publication once the following very minor issues have been addressed:

1. p. 4 ll. 105-106: "The following mappings [to] access to the data structure ..." lacks a "to" or needs to be rephrased otherwise.

2. p. 5, l. 108: "... are mathematical representations of unitary operators ..." is unclear since the term "mathematical representation" can be everything and nothing, e.g., a representation of the unitary operator relative to a specific basis, but this is not what is meant here. I think that lines 106-108 can be safely removed because eq. (12) gives a mathematically sound definition of U_P and U_Q and eq. (13) does so for P and Q.

3. p. 5, l. 110: please write "It is worth noting that THE above step is avoidable for the case of the coefficient matrix A BEING Hermitian."

4. p. 9, l. 169-170: the statement "..., our scheme can reduce the time complexity of the linear system of equations with non-square dense matrix." triggers the immediate question by how much your scheme reduces the time complexity. Also see my general comment below.

5. p. 9. eq (29) would it be possible to replace the coefficient 'd' by 'b' to be consistent with the generic problem given in eq. (1)?

6. p. 13: First of all, thank you very much for taking the time and effort to include a second numerical example. The formulation "... the second-order discrete identification model MAY BE ..." is not clear to me. Isn't this the undoubted solution based on the data you have?

I also went through the comments of reviewer #1 and your response to it. I understand your decision to weaken the statement "polynomial improvement" to "superior" in the abstract and also on page 9 (removal of "achieve polynomial improvement"). However, as a reader who does not know about this fact I would be very sceptical when reading such vague statements. Would it be possible to sketch the idea why (up to) polynomial improvements are thinkable?

Author Response

We would like to express our gratitude to you for your excellent work and constructive suggestions to improve the quality of the manuscript.
